# Mechanisms of T-Cell Exhaustion in Pancreatic Cancer

**DOI:** 10.3390/cancers12082274

**Published:** 2020-08-14

**Authors:** Didem Saka, Muazzez Gökalp, Betül Piyade, Nedim Can Cevik, Elif Arik Sever, Derya Unutmaz, Güralp O. Ceyhan, Ihsan Ekin Demir, Hande Asimgil

**Affiliations:** 1Department of General Surgery, HPB-Unit, School of Medicine, Acibadem Mehmet Ali Aydinlar University, Istanbul 34684, Turkey; didem.saka@live.acibadem.edu.tr (D.S.); Muazzez.Gokalp@acibadem.edu.tr (M.G.); betul.piyade@live.acibadem.edu.tr (B.P.); Nedim.Cevik@acibadem.edu.tr (N.C.C.); Elif.Sever@acibadem.edu.tr (E.A.S.); hande.asimgil@acibadem.edu.tr (H.A.); 2Jackson Laboratory of Genomic Medicine, Farmington, CT 06032, USA; derya.unutmaz@jax.org; 3Department of Surgery, Klinikum Rechts der Isar, Technical University of Munich, 81675 Munich, Germany

**Keywords:** pancreatic ductal adenocarcinoma, PDAC, T-cell exhaustion, epigenetics, Thymocyte selection-associated high mobility group box protein, TOXs, tumor microenvironment, TME

## Abstract

T-cell exhaustion is a phenomenon that represents the dysfunctional state of T cells in chronic infections and cancer and is closely associated with poor prognosis in many cancers. The endogenous T-cell immunity and genetically edited cell therapies (CAR-T) failed to prevent tumor immune evasion. The effector T-cell activity is perturbed by an imbalance between inhibitory and stimulatory signals causing a reprogramming in metabolism and the high levels of multiple inhibitory receptors like programmed cell death protein-1 (PD-1), cytotoxic T-lymphocyte-associated protein 4 (CTLA-4), T cell immunoglobulin and mucin domain-containing protein 3 (TIM-3), and Lymphocyte-activation gene 3 (Lag-3). Despite the efforts to neutralize inhibitory receptors by a single agent or combinatorial immune checkpoint inhibitors to boost effector function, PDAC remains unresponsive to these therapies, suggesting that multiple molecular mechanisms play a role in stimulating the exhaustion state of tumor-infiltrating T cells. Recent studies utilizing transcriptomics, mass cytometry, and epigenomics revealed a critical role of Thymocyte selection-associated high mobility group box protein (TOX) genes and TOX-associated pathways, driving T-cell exhaustion in chronic infection and cancer. Here, we will review recently defined molecular, genetic, and cellular factors that drive T-cell exhaustion in PDAC. We will also discuss the effects of available immune checkpoint inhibitors and the latest clinical trials targeting various molecular factors mediating T-cell exhaustion in PDAC.

## 1. Introduction

Pancreatic ductal adenocarcinoma (PDAC) is one of the deadliest malignancies with a five-year survival rate of only 9%. Globally, the mortality numbers are very close to incidence numbers projecting pancreatic cancer as the 7th leading cause of cancer-related deaths. Globocan statistics predict the incidence number to be almost doubled by 2040 (http://globocan.iarc.fr/) [1].

The poor prognosis associated with the lack of efficient treatment modalities makes PDAC one of the most lethal cancers [2]. PDAC tumors are unresponsive or mildly responsive to chemotherapy, radiotherapy, and immunotherapy. The desmoplastic dense stroma [3], bearing relatively low mutational loads, the low number of tumor neoantigens [4,5], the poor tumor immunogenicity [6,7], acquired tumor intrinsic therapy resistance, genetic and epigenetic instabilities, and the unique immunosuppressive tumor microenvironment (TME) are the proposed characteristics for the impaired drug delivery and low therapy response.

Highly complex pancreatic TME modulates the infiltration of immunosuppressive cells and the activity of immune regulatory molecules (Figure 1); thus, it contributes to the downregulation or dysfunctionality of antitumor immune response, including the exhaustion of T lymphocytes [8]. 

First defined by viral immunologists, T-cell exhaustion is a differentiation state of T cells upon chronic antigen exposure, which triggers T-cell receptor (TCR) signaling during chronic infections [9,10,11] and increases during aging [12].

It is also associated with tumor progression in the context of cancer. Growing pieces of evidence suggest that T cells that have undergone productive initial activation, diverge into two subtypes: (1) progenitor/memory-like and (2) terminally differentiated exhausted T cells (Tex). The latter differentiates itself from effector and memory T cells by its unique epigenetic and transcriptional program [13]. It appears that Tex cells present some characteristic features, which are (i) upregulated expression of checkpoint inhibitory receptors, (ii) decreased production of antitumor cytokines, (iii) increased secretion of tumor-promoting chemokines and (iv) high apoptosis rate [14,15]. Nevertheless, some specific stimuli, the properties of TME, the type of the tumor, and the antigen exposure mode, shape the generation of distinctive molecular and immunophenotypic features of Tex in the context of cancer. 

The exhaustion is a gradually progressing mechanism that includes distinct dysfunctional states [9]. Recent findings and therapeutic reactivation studies suggest that there is a potential therapeutic window in the formation of Tex population in which Tex are still able to proliferate and express a broad spectrum of effector function-related genes [16]. Therefore, the exploration of signaling pathways driving exhaustion in cancer to fight with immunotherapy resistant solid tumors, like PDAC, is crucial.

Here, we focus on the recent advances in transcriptional and epigenetic reprogramming mechanisms of T-cell exhaustion driven by immune modulatory signals in the tumor microenvironment of pancreatic cancer. Also, we review and discuss new emerging targets in PDAC immunotherapy and the relevant clinical trials.

## 2. Influence of the Pancreatic Tumor Microenvironment on the Function of T Lymphocytes

To evade immune surveillance, cancer cells develop an immunosuppressive microenvironment by recruiting immune suppressive cells and exert an epigenetic, transcriptomic and metabolic reprogramming in Teff lymphocytes via either secreted soluble molecules or by the expression of membranous proteins (e.g., immune checkpoints (ICP)/inhibitory receptors (IR)). Notably, the TME of PDAC comprises cancer-associated fibroblasts, a high number of immunosuppressive cells, pancreatic stellate cells (PSC), endothelial cells, neuronal network, and immune regulatory soluble factors, all together are called desmoplastic stroma [17,18,19,20]. In fact, the pancreatic tumor’s stroma contains low to moderate levels of immune infiltrates compared to the core of the tumor, which is mostly the case in melanoma. In metastatic PDAC, total T-cell infiltration is even more reduced than primary PDACs [7]. This desmoplastic stroma, occupying 50% of the total tumor mass, not onlyforms a barrier for antitumor immune cell infiltration, but also negatively effects antitumor response, including the inhibition of T-cell activation [21]. 

### 2.1. Immunosuppressive Cells

Regulatory T cells (Tregs), tumor-associated macrophages (TAMs), myeloid-derived suppressor cells (MDSCs) and regulatory B cells (Bregs) are the primary immunosuppressive cells in PDAC’s microenvironment along with activated pancreatic stellate cells (aPSCs) and dense fibrotic stroma (Figure 1) [22,23,24,25,26]. These immunosuppressive cells are already present in preneoplastic lesions (PanINs), indicating that they may be key players in tumor initiation and progression by blocking the antitumoral activity of effector CD4+ and CD8+ T cells [27]. Moreover, the delicate balance between the populations of CD4+ and CD8+ subsets determines the anti- or the protumorigenic environment. Notably, the orchestration of naïve CD4+ T cells’ differentiation into Th1, Th2, Th17, Th9, Th22, and Tregs is crucial to remove the immunosuppressive constrains from the tumor environment and to boost effector T-cell activity. As such, the dynamic ratio of Treg/Th17 determines tumor response of the immune system [28,29].

Tregs are responsible for preventing excessive or unwanted T-cell activation, maintaining self-tolerance as a defense against autoimmunity, and often correlating with cancer progression [30,31]. Typically, infiltrated Foxp3+ Tregs in TME exert suppression on effector function by secreting inhibitory cytokines, IL-10 and TGF-β or through the cell-mediated engagement of inhibitory receptors (IRs), T cell immunoreceptor with Ig and ITIM domains (TIGIT), cytotoxic T-lymphocyte-associated protein 4 (CTLA-4), programmed cell death protein-1 (PD-1), T cell immunoglobulin and mucin domain-containing protein 3 (TIM-3) [32,33,34] and promotes the exhaustion-associated transcriptomic machinery in tumor infiltrated lymphocytes. 

In the pancreatic TME, Tregs constitute almost 25% of CD4+ TILs and contribute to the increased immunosuppression. The important role of Tregs in PDAC has been shown in a murine model, in which disruption of these cells was correlated with tumor regression [35]. Tregs also elevate kynurenine concentration as a result of tryptophan catabolism by producing indoleamine 2-3 deoxygenase (IDO) (Figure 1) and lower available tryptophan in TME, which is necessary for active Teff metabolism. [21,36]. Indeed, either depletion of Tregs or blocking TGF-β signaling in tumor models in mice prevented immunosuppression of tumor-infiltrating CD8+ cells [37].

An effector CD4+ T-helper cell subset, which secretes the cytokine IL-17 and called Th17 cells, are found in human tumors (Figure 1) [28,38]. IL-17 is a potent cytokine that induces the stimulation of IL-6, TNF, G-CSF, chemokines, and matrix metalloproteases to induce inflammation [39]. Despite their vital function in host defense against pathogens, the role of IL-17 and Th17 in carcinogenesis is still controversial. The pro- or antitumorigenic function of Th17 cells is dependent on various factors, including the type of cancer, the type, and strength of the stimulation in which the cells are exposed during activation.

In the context of T-cell exhaustion, which drives immune evasion of PDAC, the plasticity of Th17 to Treg shift plays a significant role in maintaining the immunosuppressive environment. PDAC patients were shown to carry Treg dominated Treg/Th17 cell ratio [40]. Consistent with the protumorigenic effect, oncogenic KrasG12D-dependent Th17 infiltration into PanIN lesions promoted PDAC initiation and progression via IL-17 secretion of immune cells and upregulation of IL-17 receptors in epithelial cell in a murine model of PanIN harboring tamoxifen-inducible oncogenic Kras allele (Mist1-CreERT2/+; LSL-KrasG12D; R26mTmG) [41]. Also, the distribution of Th17/IL-17+ cells in patients with metastatic pancreatic cancer showed an association with higher Th17 presence in the TME and peripheral blood [42]. In PDAC, IL-17B/IL-17RB family promotes malignancy by inducing pro-inflammatory pathways and facilitating pancreatic cancer cell recruitment of macrophages. Clinical findings of IL-17RB upregulation in PDAC patients as well as the positive correlation of IL-17 signaling blockade with tumor regression in mice provide additional support for the protumorigenic effect of IL-17 on pancreatic cancer prognosis [43]. On the other hand, higher Th17 differentiation and IL-17 production were found positively associated with antitumor immunity in some cancers, including PDAC [28,38,44]. As such, Th17 tumor infiltration into IL-6-expressed murine PDAC tumor delayed tumor growth and improved survival due to Treg/Th17 balance shifted towards Th17, suggesting that IL-6 promotes this shift in TGF-β-rich pancreatic TME [45].

M2 type anti-inflammatory macrophages called TAMs also play a significant role in pancreatic tumor progression and metastasis by facilitating immunosuppressive environment for antitumor T-cells activity and proliferation through induction of immunosuppressive cytokines and enhancing the immunosuppressive capacity and the number of tumor stem-like cells in PDAC [46]. In general, TAMs, once activated by Th2 cytokines, use many strategies to induce immunosuppression. They secrete suppressive cytokines and factors, IL-10, IL-35, and TGF-β, which contribute to the impairment of Teff proliferation and activity [47]. Alternatively, TAMs can induce exhaustion by inducing PD-L1 expression on monocytes, which bind to PD-1 on CD8+ cells. Besides, they can also inhibit Teff activity by producing enzymes that deplete certain amino acids in the environment needed for Teff metabolism. As such, the overexpression of CD73 and CD39 ectoenzymes by TAMs generate pericellular adenosine and cause suppression of Teff via activation of the adenosine A2A receptor and eventually cause apoptosis [48] (Figure 2). Therefore, the modulation of TAMs has been of great interest in recent years to overcome exhaustion and dysfunction of T cells and to achieve significant antitumor responses in therapies. Zhu et al., showed that the blockade of CSF/CSFR1 signaling significantly decreased the number of tumor-infiltrating TAMs and led to the reprogramming of TAMs, which produce less immunosuppressive and more antitumorigenic factors. Interestingly, CSF1/CSFR1 blockade achieved up to 85% tumor regression in a murine model when combined with PD1/CTLA4 inhibitors and gemcitabine, improved tumor regression in this murine model as well as increased effector CD8+ and CD4+ TIL infiltration and activity [49].

PDAC utilizes multiple mechanisms to drive T-cell exhaustion. One of the predominant mechanisms is the abnormal accumulation of immature myeloid cells in the tumor due to tumor-driven changes in myelopoiesis [50,51]. In PDAC tumors, MDSCs occupy 15–20% of infiltrating cellular mass, while tumor-associated macrophages (TAMs) hold 5–10%. They are recruited to TME by tumor-driven immunoregulatory factors. MDSCs are immature myeloid cells, which suppress antitumoral immunity, leading to cancer progression. There are two major types: (1) the predominant one in PDAC cells: Polymorphonuclear (PMN-MDSCs) and (2) mononuclear (M-MDSCs) [52]. A high concentration of MDSCs indirectly leads to suppressed antigen-specific T-cell responses. It is shown that depletion of a single myeloid subset, the granulocyte-like MDSCs (G-MDSC), can unmask an endogenous T-cell response, revealing an unexpected latent immunity in GEMM of PDAC [53]. In an autochthonous mouse model of PDAC (Pdx-Cre1; LSL-KrasG12D; p53R172H), GM-CSF and CCL2 were shown to mediate cytotoxic T-cell (Tc) dysfunction through prominent infiltration of suppressive myeloid cells expressing Gr-1+ CD11b+ (Figure 1) [54]. Consistently, Gr-1+ CD11b+ infiltration was positively correlated with increased GM-CSF production in human PDAC tumors. Alternatively, and myeloid cells’ intratumoral presence are suggested to exert an immunosuppressive effect in PDAC [55]. The disruption of the crosstalk between tumor cells and TAMs due to CSF1 and BAG3 depletion in an orthotopic PDAC tumor model enhanced Tc infiltration and activation (Figure 1), proving the importance of those soluble factors [46,56]. Zhu et al., define two main subsets of macrophages in PDAC, (1) monocyte-derived and (2) tissue-resident TAMs. Tissue-resident TAMs not only persisted but undergo significant expansion during PDAC progression. They also showed that tissue-resident TAMs are more important for progression than monocyte-derived TAMs since having higher pro-fibrotic profile and their depletion significantly reduced tumor progression [57]. Similarly, the increased infiltration of both cell types is associated with poor prognosis in PDAC patients [6,58]. Once enriched in TME, soluble factors like CCL22, PGE2, and TGF-β secreted by TAMs augment further immune suppression. While CCL22 enhances Treg activation, the latter two leads to attenuating T lymphocyte function [54,59,60].

A subset of B cells called Bregs is demonstrated to have immunoregulatory functions through secretion of tolerogenic cytokines such as TGF-β and IL-10 [61]. Guo et al. detected a high IL-18 level in pancreatic cancer patients [62]. Furthermore, IL-18 is found to be responsible for the immunosuppression and decreased Teff activity in pancreatic cancer via inducing Breg proliferation, which then leads upregulation of PD-1 receptor in B cells [63].

### 2.2. Pancreatic Stellate Cells (PSCs) 

The desmoplastic stroma of pancreatic cancer is mainly comprised of activated PSCs (aPSCs) and myofibroblasts. PSCs are activated by cytokines, including TGF-β, TNF-α, IL-1, and IL-6. They produce components of the extracellular matrix, i.e., laminin, collagen, and fibronectin, and MMPs, which give rise to pancreatic fibrosis [64]. They present certain markers such as vimentin, glial fibrillary acidic protein (GFAP), and α-smooth muscle actin (α-SMA), fibroblast activating proteinase which promote the progression of non- invasive PanIN lesions to invasive PDAC. aPSCs exert their tumor-promoting effect by recruiting suppressive subtypes of immune cells in the stroma by secreting IL-6 M-CSF [65]. In particular, they promote the differentiation [65], recruitment and the proliferation of MDSCs, as well as IL-35-secreting Bregs in the hypoxic PDAC microenvironment (Figure 1) [66,67,68]. They are impairing antitumor T-cell function by inducing immune checkpoints on Tc cells in a CXCL12-dependent manner via secretion of IL-6 (Figure 1). Also, Ene-Obong et al. reported that aPSCs sequester antitumor CD8+ T cells around nonadjacent regions in the stroma, resulting in low infiltration of CD8+ cells into the primary tumor epithelial cells in KPC (Pdx-1-Cre; LSL-KrasG12D/+; LSL-Trp53R172H/+) mice, which is associated with shorter survival in human PDAC [69]. Despite the protumor role of aPSCs in PDAC, there are also contradictory findings showing aPSCs’ antitumor function [19,70]. Thus, the exact role of stroma in PDAC progression and immune suppression remains ambiguous. In their study, Ozdemir et al. showed that deficiency of α-SMA+ myofibroblasts in stroma resulted in concentrated Treg and decreased Teff in PKT (Ptf1a-Cre; LSL-KrasG12D; Tgf-βr2flox/flox) mice which were corroborated by the human data that PDAC with low number of myofibroblast was associated with shorter survival [19].

### 2.3. Amino Acids

T-cell activation is also regulated by the presence of soluble immunomodulatory factors in TME. Tryptophan, and l-arginine are required for effector function and T-cell survival [71,72]. However, intratumoral myeloid cells and Tregs, as well as cancer cells, secrete a high amount of specific enzymes, causing the breakdown of these essential amino acids and their depletion [21,73].

IDO, which catalyzes tryptophan to kynurenine conversion, is overexpressed in PDAC [74], contributing to attenuated antitumor T-cell responses. Furthermore, increased kynurenine/tryptophan ratio in the serum of cancer patients who showed resistance to PD-1 blockade implicates that IDO should also be targeted in combinatorial immunotherapies [74,75].

l-arginine is necessary for Tc function, and L-arginine’s high intracellular levels improved T-cell proliferation and antitumor function in mice [72]. On the other hand, elevated arginase 1 (ARG1) produced mostly by MDSCs decreases l-arginine availability, thus decreasing the effector function in many cancers [76,77,78]. Likewise, high expression levels of ARG1 is associated with suppressed T-cell responses and shorter overall survival of metastatic PDAC patients [79,80,81]. Nevertheless, the pro- and antitumor roles of l-arginine remains contradictory, especially for arginine auxotrophic tumors. Some studies showed that arginine deprivation in head and neck cancers and in pancreatic cancer cell lines leads to impairment of metastatic ability and cell death [82,83,84].

Adenosine and adenosinergic signaling support Tregand inhibit the Teff function. The hyperactive signaling was shown in various cancers and mostly correlated with poor prognosis in cancer patients [85,86]. In PDAC, apoptotic Treg in the hypoxic TME was shown to express a high amount of the CD73 enzyme that converts adenosine triphosphate to adenosine, contributing to the inhibition of cytokine expression of Teff (Figure 1). In a xenograft nude mouse model, knockdown of CD73 resulted in slow tumor growth and increased sensitivity to gemcitabine [87]. The authors also showed that human PDAC tissues express attenuated levels of miR-30a-5p, which regulates CD73 protein level, suggesting that tumor growth can be inhibited by elevating miR-30a-5p levels by gene therapy to relieve immunosuppressive conditions and improve gemcitabine sensitivity [87]. Another critical factor correlated with PDAC progression is focal adhesion kinase (FAK). Hyperactive FAK in the tumor was associated with high fibrosis and reduced infiltration of Tc, indicating its indirect function in forming of dysfunctional T cells and PDAC progression [88].

## 3. Inhibitory Receptors

The upregulation of inhibitory receptors (IRs) is described as the hallmark of T-cell exhaustion upon chronic infections and cancer. Immunosuppressive cytokines/factors and cells in the TME in the presence of persistent tumor antigen stimulation induce prolonged and increased expression of cell surface IRs including, CTLAntigen-4 (CTLA-4), anti-programmed cell death-1 (PD-1), Mucin-3/T-cell immunoglobulin (TIM-3), T-cell activation gene (LAG-3) and T-cell tyrosine-based inhibitory motif (ITIM) on TILs [36] (Figure 2). Upon binding to their cognate ligands on cancer cells, T-cells’ effector function and proliferation are gradually reduced. 

Given that their blockade can partially reverse the partially exhausted phenotype, the expression profile and the levels of IRs are also drivers of cancer-related T-cell exhaustion process [89]. In PDAC, like in the other cancers [90,91,92,93], IRs prevent CD4+ and CD8+ TILs from being effectively functional; however, they might also play a role in the regulation of T-cell infiltration in pancreatic tumors. Here we discuss IRs role in T-cell immunity in PDAC.

### 3.1. CTLA-4 and PD-1/PDL-1 

Unfortunately, neither single nor combined anti-CTLA-4 immunotherapy trials became successful so far in treating PDAC patients [94,95]. Thus, there is still a lack of knowledge to be filled about molecular mechanisms underlying such unresponsiveness of pancreatic cancer. To enlighten this, Bengsch et al. used the KPC mouse model and found that CTLA-4 blockade on Tregs accumulated in peritumoral lymph nodes and on Teff cells enhanced CD4+ infiltration; however, it was not sufficient to recruit CD8+ cells into the TME (Figure 2) [96].

Programmed cell death-1 (PD-1), mostly expressed on effector CD4+ Th cells and CD8+ TILs [90,97,98,99], binds to its ligands, PD-L1 and PD-L2, on solid tumors [100], on tumor-infiltrating dendritic cells [101], and on tumor associated-macrophages and MDSCs [102], to prevent chronic activation of T cells [15] (Figure 2). If antigen-overexposure occurs, PD-1/PD-L1 signaling creates a positive feedback loop where this signaling becomes dominant and generates an exhausted T-cell population within the tumor and its periphery by inhibiting T-cell activation upon the recruitment of SHP2 tyrosine phosphatase which dephosphorylates CD28, attenuating TCR signaling [103]. PDAC is described as ‘immunologically cold’ compared to highly immunogenic melanoma because of very low surface presentation of neoantigens, and the insufficient Tc infiltration into the tumor core because of fibrotic trap and TAMs localized in the surrounding of tumor [3,55,102,104] which result in poor clinical outcomes from immune-checkpoint inhibitors targeting PD-1/PD-L1 and CTLA-4 [70,105,106]. A comprehensive retrospective study on resected PDAC tumors reported four major subclasses of tumors based on genomic, transcriptomic and, clinicopathological data. High levels of tumor neoantigens exist in the subtypes with impaired double-strand break and mismatched repair mechanisms, implicating that immunotherapy can be successful if applied to the right patient [107]. Fortunately, recent advances in genomics and transcriptomics have been discovering new target proteins that can improve tumor regression when combined with existing therapies for PDAC [108].

### 3.2. LAG-3

LAG-3 exerts differential inhibitory effects on TILs by cooperating with other co-inhibitory molecules upon the MHC II association (Figure 2) [109]. Its excessive expression leads to dampened CD4+ T-cell activation, enhanced Treg suppressor activity, and decreased cytotoxic function of Tc [110]. Elevated expression of LAG-3 on TILs from patients with PDAC was detected along with increased PD-1 and CTLA-4 [111], implicating that dual and triple blockade of such inhibitory receptors might improve the effectiveness of immunotherapy treatment of PDAC. We will discuss such modalities and the efficacy of multiple blockades of IRs in the immunotherapy section below [112,113].

### 3.3. Galectin Family

The role of deregulated expressions of Galectins family proteins is implicated in tumor progression and tumor immune evasion in many cancers. They mediate the crosstalk of tumors and TME [114]. There is increasing evidence that Gal1, Gal3, and Gal9 play important roles in stromal modulation of ECM, T-cell infiltration, activation, apoptosis, and the formation of the immunosuppressive environment in PDAC in humans [115]. Upregulation of Gal1 was detected in pancreatic tumors, which activates PSCs, thereby promoting fibrosis in stroma via autocrine signaling [114,116]. Further, in vitro studies showed that the paracrine signaling of Gal1 enhances tumor cell proliferation, invasion, and migration, while it induces apoptosis of T lymphocytes and proinflammatory cytokine secretion [117]. On the contrary, the absence Gal1 gene in oncogenic KrasG12D-driven PDAC tumor in mice retained an increased number of CD3+, CD4+, CD8+ T lymphocytes, and decreased levels of CD11b+Gr1+ MDSCs in TME.

Gal3, overexpressed by PDAC tumors in both human and mouse pancreas with oncogenic KrasG12D, is associated with tumor progression and immune modulation. Gal3 modulates T-cell function in various mechanisms. It impairs IFN-γ secretion of TILs when neutralized and removed from the T-cell surface [118]. Indeed, Gal3 interacts with immune checkpoint LAG-3, which is necessary for Gal3–mediated suppression of Tc (Figure 2). Moreover, studies in patients with GM-CSF–secreting allogeneic PDA (GVAX) and PDAC mouse models indicate that Gal3 modulates plasmacytoid dendritic cells, which are the potent activator of Tc cells and development of MDSCs [119].

Finally, the role of the Gal9, expressed in both leukocytes and tumor cells in PDAC, has been shown. The blockade of Gal9/Dectin-1 interaction improved intratumoral T-cell activation in PDAC and associated with TAM reprogramming, while only Gal9 inhibition enhanced CAR-T-cell cytotoxicity and alleviated PDAC immunotherapy resistance [120,121]. Gal9 interaction with co-inhibitory receptor TIM-3 on Teff, Th, and innate immune cells induce dysfunctional programming in T-cells in chronic infection [122]. Whereas the single nucleotide polymorphism in ORF of TIM-3 in the Chinese population was shown to increase the susceptibility to gastric, non-small lung cancer and pancreatic cancer, implicating the importance of Gal9/TIM-3 signaling [123,124].

### 3.4. TIGIT

TIGIT is a recently identified member of the CD28 family, acting as a co-inhibitory receptor (Figure 2) [125]. It expresses on NK cells and T cells, specifically on activated memory and follicular Th cells, and a subset of Treg cells [125]. In T cells, TIGIT binding inhibits T-cell proliferation, cytokine production, and TCR signaling in a cell-internal manner [126,127].

### 3.5. SLAMF6

Signaling lymphocyte activation molecule family 6 (SLAMF6) is a cell surface receptor expressed on activated T lymphocytes [128], macrophages, and APCs [129]. Although its co-regulative function on antigen-driven T-cell response was shown in viral infections, its immunomodulatory role in cancer is not clear. Yigit et al. examined Tc cells of anti-SLAMF6 injected mice with melanoma to test whether there is an increase in effector functions. Intracellular staining of Tc cells showed that effector markers, lysosomal CD107a and granzyme B, and IL-2 expressing Tc cells were significantly increased in number. A lower percentage of CD8+PD-1+ TILs in anti-SLAMF6 injected mice were found compared to the control group, which suggests the activation of Tc’s in the tumor [130]. There is also a relation between the SLAMF6 gene and PDAC. The analysis of miRNAs from pancreatic tissues of 178 PDAC patients and four healthy subjects showed that the SLAMF6 gene was predicted to be regulated by significantly under-expressed miRNAs in PDAC [131]. Further investigations are needed for clarifying the role of the SLAMF6 in the immune modulation of pancreatic cancer. 

### 3.6. VISTA

V-domain Ig Suppressor of T-cell Activation (VISTA) has recently drawn attention as a potential target for PDAC. Blando et al. showed that VISTA predominantly exists in the pancreatic stroma of human metastatic PDAC patients and reciprocally correlated with antitumor T-cell response and cytokine production of TILs (Figure 2) [7]. Given the expression profiles of Tc from human PDAC patients, Balli et al. suggested that patients with pancreatic cancer can be categorized to apply patient-specific treatment modalities based on co-expression of CTLA-4, TIGIT, TIM-3 and VISTA (Figure 2) [132]. In fact, combinatorial therapies, including VISTA for tumors with highly immunosuppressive TME like PDAC, can be promising since VISTA is induced by hypoxia [133] and is mostly presented on MDSCs, TAMs and T cells [134]. Therefore, relieving immunosuppressive factors from TME can enhance the recovery of exhausted T cells.

### 3.7. TIM-3

T-cell immunoglobulin and mucin domain 3 (TIM3) belongs to IRs containing non-conventional signaling domain without a defined inhibitory domain in the cytoplasmic tail and relays inhibitory signaling through interaction with multiple ligands, conferring a context-specific Tex relevant activity [135]. It is mostly expressed in inflammatory IFNγ-producing CD4+ T cells (Th1), CD8+ T cells, NK cells, and tissue-resident FoxP3+ Treg [136]. Exhausted T-cell populations with a severe phenotype in both cancer and chronic infections are shown to co-express PD-1 and TIM-3 [137]. Nevertheless, in solid tumors, they can comprise the majority of TILs, leading to a failure in tumor regression [138]. In metastatic gastrointestinal solid tumors with ascites, including pancreatic cancer, TIM-3 co-expresses with PD-1 on TILs, which is associated with worse clinical outcomes [139]. According to Pu-Ji et al., high TIM-3 expression was significantly higher in pancreatic cancer than in healthy pancreas based on immunohistochemical analysis of patient samples. Also, a significantly shorter median survival of patients with TIM-3 expression than the patients with negative TIM-3 suggests that TIM-3 plays a role in immune infiltration, evasion, and metastasis of pancreatic cancer [140].

## 4. Transcriptional and Epigenetic Reprogramming of T-Cell Exhaustion TOX and TOX2

In recent years, new regulatory mechanisms and the associated genes that drive progressive differentiation of effector to dysfunctional T cells in chronic infection and cancer were discovered by a group of researchers utilizing transcriptomics, epigenomics guided mass cytometry profiling, and systematic gene set analysis [13,141,142,143,144]. In their cutting-edge paper of 2018, Bengsch et al. identified exhaustion-specific gene signatures and they set them as biomarkers for exhaustion, including IRs, metabolic enzymes, chemokine, and cytokines transcription factors in chronic infection [13]. Among those, Tox genes stood out as their expression seemed unique to exhausted phenotype and was not detected in naïve, memory, and effector subsets. Further mechanistic studies came up a year later in subsequent publications by Khan et al., Scott, Alfei et al., Seo et al. [141,142,143,144]. They revealed the function of TOX and TOX2 as central regulators of exhaustion. By utilizing ATAC-seq, RNA-seq, sc-RNA-seq, the studies revealed the signaling and epigenetic regulators of T-cell exhaustion in chronic infections and cancer. The common finding of all the groups was that TOX regulated by NFAT1 was distinctively expressed at very high levels in tumor-specific exhausted T cells in an inducible liver carcinoma mouse model called AST mice which bears Cre-mediated expression of the oncogene SV40 T antigen (Tag), the Albumin-floxstop-Tag (AST) mouse model generated by Stahl et al. to study antitumor immunity [145]. Both transcriptional and epigenetic reprogramming by TOXs alter the gene expressions of IRs; Teff related transcription factors (TFs), and cytokines/cytotoxic molecules in T cells (Figure 2).

Scott et al. founded their recent study on their previous findings [14] that antigen-specific naïve cytotoxic T cells carrying oncogene SV40-T antigen (TCRtag) transform into dysfunctional T cells driven by epigenetic changes during tumorigenesis. The authors showed that there were progressively increasing and persistent expression of TOX in Tc liver cancer and murine melanoma during tumor progression, in contrast to temporary upregulation of TOX in memory and effector cells in acute infection [141]. Moreover, the TOX expressing tumor-specific CD8+ cells demonstrated phenotypic exhaustion markers, i.e., high levels of IRs, low expression of effector cytokines IFN-γ and TNF-α; however, low levels of TCF-1, a key transcription factor that determines exhausted T cell fate for cellular differentiation and persistence [141,146]. Furthermore, they presented more proof that tumor-specific T-cell exhaustion is driven by a prolonged encounter with tumor-specific antigen as tumor non-specific Tc did not express a high level of TOX gene and remained functional, unlike the tumor-specific TCRtag cells. Lastly, a significant finding of this study was that T cells engineered to knock out TOX in T cells (TOXKO TCRtag) transferred to mice with tumors were far from having strong effector function. They produced low Granzyme B, IFN-γ, and TNF-α and showed low levels of IRs (PD-1, TIM-3, LAG-3, 2B4) presentation, which implies that modulation of IRs might be impaired with functionality. Eventually, they decreased in number and died, which was corroborated by increased levels of apoptosis-associated molecules, active caspases 3 and 7, Annexin V. The analysis of human Tc from melanoma, breast, lung, and melanoma cancer supported the findings in mouse experiments that the exhausted T cells are a tumor-reactive, TOXhigh, IRhigh distinct population [141,147].

A detailed molecular mechanistic explanation is provided in Alfei et al.’s results in the lymphocytic choriomeningitis virus (LCMV) mice model and the hepatitis C virus in humans [144]. Previously, it was shown that the chromatin of Tex cells remodel upon encounter with tumor antigens to a transient open state and remain open and stable for long-term due to continuous exposure, and keep its open state even after the chronic antigen stimulation was resolved [14,148]., Notably, the state of chromatin of Tex can be used to quantify reversible and irreversible Tex populations in tumors based on differentially expressed membrane proteins such as CD38, CD39, 2B4, and CD101. The study of Alfei et al., elucidated this mechanism in chronic infection mouse model and showed that initial TOX induction is resulted from the demethylated Tox locus which is initially induced by high antigen stimulation of the T cell receptor. The authors found that the chromatin regions expressing the cytokine transcription factors, inhibitory receptors, effector marker proteinIL-10, IFN-γ, and TNF-α, NR4a2, NFAT pathway were more accessible and transcriptional accessibility differentially changed in CD8+ based on TIM3 expression in the course of chronic infection [5]. In this regard, TOX function in reprogramming T cells gains prominence in PDAC since a high population of Eomes+ PD-1+ Tc is associated with low antitumor immunity [149] and short survival in PDAC patients [58].

Interestingly, when T cells were engineered to have a conditional deletion of the DNA-binding domain in the Tox gene (mutant-TOX), in chronic infection, PD-1 expression decreased, cytokine production induced, effector KLRG1+, and viral control improved, indicating more a polyfunctional, more-effector phenotype development [144]. As also observed by Scott et al., mutant-TOX cells showed an initial expansion but then died in the long term under chronic antigen stimulation, despite they, in the short term, expressed the same level of the transcription factor TCF-1 for T-cell maintenance compared with WT. Overall, these findings indicate that TOX serves as a supporting factor for the tumor antigen-specific Tc to persist in the tumor environment and a self-protection mechanism from overstimulation and dying. 

Khan et al. explored early epigenetic events mediated by TOX and TOX interacting proteins that cause a shift from effector to an exhausted state in the Tc population in acute and chronic infections and tumor progression. Their principal findings point out that TOX makes protein complexes with chromatin modifiers for the chromatin openings and the closings. Among the binding partners, the HBO1 complex, which is involved in acetylation of histone of H3-H4, was identified as major binding partners of TOX. Diversely, it was also found that TOX binds to repressive chromatin modifiers such as DNMT1, LEO1, PAF1, SAP130, and SIN3A. Thereby, KLRG1+ Teff cell differentiation was suppressed, cytokine production and cytotoxicity were lost. Consequently, the whole transcriptome in TILs was subjected to change due to either direct or indirect effects of TOX-driven chromatin reprogramming. For instance, certain chromatin regions holding Nr1d2, Atf3, Bcl6, Sox4 transcription factors, and cellular stemness-related genes Nanog, Sox2 were blocked in the absence of TOX, implicating TOX has a wide range of binding partners to regulate open and close chromatin states [142].

Chimeric antigen receptor (CAR)-T cell therapy, which utilizes genetic engineering to redirect a patient’s T cells to target cancer cells, showed promising results in hematological malignancies but limited function in solid tumors such as pancreatic cancer [150]. Beside many challenging side effects, CAR-T cells can also be exhausted with time and become nonfunctional [151,152,153]. To find if exhaustion occurs in engineered T cells, Seo et al. transferred CD8+ CAR T cells targeting human CD19 cells into mice with melanoma tumors (human CD19-expressing B16-OVA melanoma) to analyze the expression profile of the CAR-Tumor Infiltrating Lymphocytes (CAR-TILs). CAR-TILs expressed gradually increasing amounts of TOX, TOX2, PD-1, and TIM-3, while IFN-γ and TNF level diminished, implicating that the engineered T cells become exhausted over time. Also, CAR-TILs with double deficiency of TOX and TOX2 were more effective in mediating tumor regression than single knockouts or WT. Then, they compared TOX double knockout (ToxDKO) with WT CAR-TILs. They showed that expression of PD-1, TIM-3, LAG-3, was significantly lower and their cytolytic activity outperformed their wild type counterparts [143]. The lower TCF-1 and Eomes levels in TOXDKO suggest that TOXs regulate the CAR-TILs’ fate in a tumor. The researchers also showed a mechanistic link for the upregulation of TOX, TOX2, and the nuclear receptor NR4A1, which is identified to induce T-cell dysfunction [154]. As shown by Khan et al., calcium/calcineurin signaling activates transcription factor NFAT in CD8+PD-1highTIM3high CAR-TILs. TOX and NR4A generate a positive feedback loop, and with the contribution of NFAT, they all drive the upregulation of IRs in CAR-TILs [142].

Overexpression of vascular endothelial growth factor (VEGF) and VEGF receptors played an essential role in the formation of high microvascular density in pancreatic cancer [155,156,157]. They emerged as adverse prognostic factors in terms of patient survival [158]. A recent study by Kim et al., points out a VEGF-A induced TOX signaling cascade, which drives transcriptional reprogramming for T-cell exhaustion in anti-PD-1 resistant colorectal cancer [159]. In the presence of VEGF-A, anti-CD3 stimulated T cells from healthy human subjects presented upregulation in inhibitory receptors and the expression of proapoptotic molecules, indicating the critical role of VEGF-A in TOX-regulated epigenetic changes in chromatin of T cells that result in attenuated T cell effector capacity [160].

### 4.1. NFAT and NR4A and TOX-Associated Transcription Factors

Nuclear Factor of Activated T cells (NFAT), activated via calcium/calcineurin pathway, has a role in the regulation of gene expression in T cells, and it is highly expressed in pancreatic cancer (Figure 2) [147,161]. The expression of TOX genes was shown to be activated by NFAT, both in vivo and in vitro [143]. The findings of Khan et al. revealed the mechanism underlying the upregulated TOX expression Tex. They showed that NFAT2 is necessary to induce TOX expression but not indispensable for induction since enforced TOX expression in NFAT2KO mice also resulted in Tex [142]. Xiao et al. also showed NFAT overexpression inhibits the Teff function by binding to the transcription factor activator protein 1 (AP-1) site on chromatin [162]. Basic Leucine Zipper ATF-Like Transcription Factor (BATF) and Interferon Regulatory Factor 4 (IRF4), are TCR signaling sensitive molecules that are important transcription factors working in collaboration with NFAT [163]. Man et al. defined these transcription factors in chronic infection. Especially NFATc1, BATF, and IRF4 all converge to establish features of exhaustion in Tc, including upregulation of IRs and decreasing the TCF1+ T cells in number (Figure 2) [163]. Given the roles, these two transcription factors should be included in mechanistic studies of T-cell exhaustion.

NR4A orphan nuclear family consisted of Nr4a, Nr4a, Nr4a3, places in the downstream of NFAT. Two papers published in 2019 by Liu et al. and Chen et al. demonstrated that this family, particularly NR4A1, was upregulated and led to dysfunctionality by modulating epigenetic and gene expression features in Tex cells chronic infection and cancer [154,164]. Thereby, LAG-3, PD-1, and TIGIT and transcriptional repressors were upregulated, while effector and metabolism-related genes were suppressed due to significantly different H3 trimethylation levels on lysine 4 (H3K4me3) at the corresponding loci [154]. The authors also interrogated NR4A1 function in a mouse model of lymphoma by deleting it (Nr4a1DKO) in CD8+ T cells and transforming them into tumor-bearing mice. They found that in the absence of NR4a1, Teff exhibited significantly better tumor infiltration and effector function at eliminating tumors than WT CD8+ T cells and low levels of PD-1 and TIM-3. Regarding mechanistic explanation, they revealed that Nr4a1 competes with c-Jun and mostly with AP-1 to bind their consensus sequences, thus induces exhaustion by antagonizing AP-1 mediated gene expression. 

Investigation of CAR-TILs by Chen et al. corroborated the findings of Liu et al. As such, Nr4A1 and NR4A2 expressions were positively correlated with the expression of PD-1 and TIM-3, and NR4A was enriched in transcriptionally active and accessible regions in CD8+PD-1high TILs from human melanoma and non-small cell lung cancer [148,164]. The authors also presented that in triple Nr4a triple knockout (NR4ATKO) CAR T cells, bZIP and Rel/NFκB binding motifs were found to be more accessible compared to wild type in mouse solid tumor models. NR4ATKO CAR-T cells exhibited better performance on inducing tumor regression and prolonged the survival of tumor-bearing mice compared to those with WT CAR-TILs and single-gene knockouts. Briefly, these results suggest that the efficacy of existing immunotherapies on solid tumors can be improved by additional targeting of NFAT and NR4A (Figure 2), whose expressions are also correlated with tumor initiation and progression in PDAC [165]

### 4.2. 4-1BB

4-1BB (TNFRSF9 or CD137) may be considered one of the immune-modulating molecules with contradictory tumor activity. Kim et al. showed that among CD8+ TILs, extracted from hepatocellular carcinoma (HCC) patients, 4-1BB+ PD-1high Tc TILs exhibit significantly higher levels of tumor reactivity than 4-1BB− PD-1high Tc [166]. Choi et al. reported that antitumor immunity is enhanced in 4-1BBDKO mice as a result of the increase in NK cells due to the shift of the dominant type of immune cells from the innate NK cell to the adaptive Tc via 4-1BB signaling [167]. It is known that constitutive activation of oncogenic KrasG12D, which approximately 90% of pancreatic cancer incidences exhibit, upregulates 4-1BB in tumor cells through MAPK and NF-κB signaling [168,169]. Therefore, many clinical CAR-T cell trials in pancreatic cancer are also using the 4-1BB receptor, targeting mesothelin, MUC1, CD19, CD133 (Table 1). However, the effect of the drifting immune cell types and the potential consequences on T-cell exhaustion should be kept in mind. 

### 4.3. STAT1

Signal Transducer and Activator of Transcription 1 (STAT1) is a transcription factor involved in the JAK-STAT signaling pathway and defined as a prognostic factor in pancreatic cancer, is inversely associated with metastasis and tumor differentiation [170,171]. Furthermore, STAT1 was found to be an inhibitor of Forkhead Box protein M1 (FoxM1) that acts as an oncogene via NF-κB signaling in pancreatic cancer [172]. Ryan et al. also reported its role for T-cell function, who indicated that complete deficiency of STAT1 in vivo showed poor expansion CD4+ and Tc expansion and decreased TNF-α production, therefore leading to increased tumor growth [173].

## 5. Metabolic Changes in T-Cell Exhaustion

Metabolic changes in T cells can be considered as a response to the increasing demand for T cell activation. However, it is also crucial for the differentiation and appropriate function of T cells, highly dependent on metabolites from energy metabolism [174]. Naive and quiescent cells demand mostly ATP-generating processes, which are met by oxidative phosphorylation [175].

The metabolic switch from oxidative phosphorylation to aerobic glycolysis takes place T cells for activation and cancer cells. Effector cytokine function of activated T cells is highly diminished if aerobic glycolysis is inhibited [176]. Glucose transport is limited in early exhaustion due to the downregulation of GLUT1 and the excessive consumption of glucose by tumor cells (Figure 2) [175,176]. Mitochondria of exhausted T cells in infections are more abundant but dysregulated, which leads to a decrease in mitochondrial function and an increase in the production of reactive oxygen species (ROS) (Figure 2) [177].

In contrast to the metabolic changes seen in the exhausted cells in infections, both mitochondrial mass, and function, diminished in TILs [178]. Tex cells during infection respond to both genetic deletions of PD-1 or blockade of the PD-L1. However, mitochondrial biomass in TILs does not respond to the blockade of PD-1, although blockade results in tumor regression [177,178].

Hypoxia is a condition that plays a well-known role in tumor progression and tumor survival in solid tumors [179]. The tumor suppressor and the negative regulator of HIF, von Hippel–Lindau (Vhl) protein, and hypoxia-inducible factor (HIFα) partly control Tc activity as a response to hypoxia. It appears that enhanced HIFα activity mediating the transcriptional profile and the differentiation of Tc cells play an important role in infection and tumor clearance [180].

## 6. Immunotherapy in Pancreatic Ductal Adenocarcinoma-Current Status

In pancreatic cancer, immunotherapy has not yet shown significant clinical activity and is mostly inefficient as monotherapy due to low immunogenicity and desmoplasia [181]. The abundant stroma of pancreatic cancer causes a hypoxic microenvironment, further leading to the recruitment of immunosuppressive cells and inhibiting antitumor immunity [21]. Tc is essential for successful tumor immune response, and tumor-specific T cell infiltration is associated with more prolonged survival in patients with tumor-associated antigen-specific Tc responses than in patients without it. However, PDAC is defined as immunologically cold, which means it has a low degree of inflammation since the fibrotic barrier seems to impede the T-cell recruitment [3,54,69,182]. Furthermore, even if cell T cells arrive in the tumor, T-cell exhaustion, and the immunosuppressive TME leads to unsuccessful immune [183]. Although the tumor microenvironment in pancreatic cancer is highly immunosuppressive, recent advances in immune-based therapies hold promise for treating this deadly disease.

Although checkpoint inhibitors (CPI) are commonly used in many other cancer types such as melanoma, non-small cell lung cancer, ovarian cancer, renal cell cancer [184], there have been no objective responses as either single agent or combination of CPIs in patients with pancreatic cancer (Figure 2) [185]. CPIs target negative immune checkpoint molecules, including PD-1, PD-L1, and CTLA-4, which causes an increased immune response and decreased tumor progression [186]. There are several reasons for the failure of immune checkpoint inhibitors in PDAC. Most PDAC display low levels of PD-1+ T cell infiltration and a small number of neoepitopes, which can be considered as the reason for the reduced efficacy of checkpoint inhibitors [187]. In a rare subset of patients with microsatellite instability-high (MSI-high) tumors, and thus a high number of neoepitopes, patients can respond to PD-1 blockade [188]. Blockade of PD-1/PD-L1 and CTLA-4 is effective up to 50% of the patients with immune sensitive cancers such as melanoma, non-small cell lung cancer, squamous cell carcinoma of head and neck cancers, renal cell carcinoma, bladder cancer, and Hodgkin’s lymphoma according to the FDA’s objective response criteria [189]. However, PDAC turns out to be one of the least responsive tumors to single-agent treatments targeting PD-1/PD-L1 and CTLA-4 signaling [186]. Given the results of anti-PD-1 treatment of progressive metastatic carcinoma with mismatch repair deficiency, which promotes somatic mutations and possibly mutated neoantigens, immune checkpoint inhibition modalities seem promising to overcome T-cell exhaustion in PDAC patients with mismatch repair deficiency [190]. Due to the limited effect of a PD-1 inhibitor in pancreatic cancer, it is shown that CXC chemokine receptor 4 (CXCR4) blockade promotes T cell infiltration with its synergistic effect with PD-1 inhibitors in mouse models [67,191]. Bockorny et al. also showed that combined CXCR4 and PD-1 inhibitors promoted an increase in T-cell infiltration and a decrease in MDSCs in pancreatic cancer patients [192].

Although PDAC is considered as unresponsive to checkpoint inhibitors, there are promising results with a combination of a T-cell inducing vaccine and a granulocyte macrophage colony-stimulating factor secreting PDA vaccine (GVAX) CTLA-4 and PD1/PDL1 blockade. With CTLA-4 inhibitor ipilimumab and GVAX, metastatic PDAC patients overall survival improved [193]. Despite the small sample size of the study, patients benefited from this combination treatment due to enhanced T-cell responses seem to benefit more likely to, implying T-cell induction and maintenance of T-cell response could be a critical step for checkpoint inhibitors. The same group investigated PD1/PDL-1 blockade combined with GVAX treatment in liver metastasis of the PDAC mouse model. Notably, they showed GVAX significantly induced PDL-1 expression on tumor and it improved the effector function of CD8+ T cells and IFN-γ production, resulting in improved survival compared to monotherapy regimens of both [194]. Due to the exhaustion of antitumor T cells, the number of T cells interacting with tumor antigens is diminished, limiting the efficacy of PD-1 blockade [195]. In order to overcome T-cell exhaustion, intratumoral in situ injection using dual CD40-TLR4 stimulation was applied and exhausted Tc cells were eliminated in murine models with bilateral tumor approach to assess its efficacy both on the treated tumor and on the distant tumor which improved tumor control with the addition of PD-1 inhibitor [196]. Activation of the CD40 receptors by tumor cells is an important step for T-cell immunity. Thus, it is worthy of mentioning that stimulating antigen-presenting cells is also a way to boost the immune responses in PDAC. CD40 agonist mAb is one of the possible targets, which has been shown to have antitumor activity in solid malignancies. [197,198]. In the case of PDAC, the clinical trial of CD40 agonist monoclonal antibody (mAb) combined with gemcitabine gave hopeful results [199]. Also, the phase 1 study CD40 agonistic mAb plus gemcitabine and nab-paclitaxel with or without nivolumab showed significant antitumor activity in PDAC patients [200]. Although these studies gave promising results, their toxicity such as cytokine release syndrome, vascular and hematologic complications, and liver toxicity should be kept in mind.

Inhibitors of TGF-β are shown effective in preclinical models [201]. Besides, a combination of TGF-β inhibitors with gemcitabine improved overall survival compared to gemcitabine alone in patients with unresectable pancreatic cancer [202].

Multiple other immune molecules can inhibit T cell responses in cancer, including TIM-3, TIGIT, and LAG-3, and these molecules should be investigated in the design of further immunotherapy modalities. Another critical issue in immunotherapy is the presence of immunogenic tumor antigens to drive a cancer-specific T-cell response. Multiple antigens have been studied previously in this context, such as telomerase, MUC1, enolase, WT1, Kras, and mesothelin [203,204]. These antigens can be potential targets for increasing the immunity of the tumor, but their clinical utility yet needs to be shown. Although current clinical trials show that PDAC is an immunologic outlier, with a better understanding of TME of PDAC and T-cell exhaustion, new solutions for patients are likely to be underway. Future combination therapies, including CPI, vaccines, and those that work against exhausted T cells, which is a significant obstacle in immunotherapy, are promising strategies. We listed all the recently used immunotherapy approaches in pancreatic cancer in Table 1.

## 7. Conclusions

Here, we reviewed the recent and most relevant studies on the immunosuppressive tumor microenvironment-induced signaling pathways, transcription factors, and epigenetic programming driving T-cell exhaustion, with a focus on pancreatic ductal adenocarcinoma. To develop new treatment modalities, all molecular factors mentioned in the review should be studied as T-cell exhaustion remains one of the main resistance factors against immunotherapy in PDAC. Since many factors lead to T-cell dysfunction, it is not easy to find a single responsible factor and to see a miracle by fixing it. It is also controversial if the exhausted state of the T cell can be reversed or not. While some studies say that the phenomenon is pharmacologically reversible, some say that there are two states of exhausted T cells, and reversal of the utterly exhausted cell state is impossible [36,143,181,205].

Recent studies showed that upon a series of genetic and epigenetic alterations in the chromatin of T cells, the genes on the NFAT–TOX axis are activated and lead to the upregulation of IRs, loss of effector function, eventually resulting in T-cell exhaustion in chronic infection and cancer.

Besides, some studies demonstrated that the complete ablation of these regulatory genes also creates dysfunctional, nonsustainable T cells. Thus, fine-tuning of T-cell activity by both inhibiting exhaustion driving molecules and in parallel targeting molecules that enhance effector function may be more successful in improving the efficacy of existing immunotherapy regimens. Furthermore, inefficient and unsuccessful clinical outcomes of immunotherapies for classical checkpoints PD-1 and CTLA-4 would be reversed or enhanced if combined with second-generation checkpoint targets, TIM-3, TIGIT, LAG-3, mostly expressed on exhausted phenotype. Therefore, these findings add to our understanding T-cells’ differentiation in the tumor microenvironment and, eventually, be used to develop new strategies to treat immune-outlier tumors like pancreatic cancer. It is generally accepted that the induction of exhaustion in T cells in chronic infection and cancer is an evolution of the immune cells. It serves as a physiologic mechanism to prevent immune overstimulation and survival of the T cells in chronic antigen encounters [141,144]. Therefore, one should be very careful when manipulating the exhausted T-cell subset. They can cause immunopathogenesis, massive edema, and autoimmune diseases if T cells are unleashed for a long time. Thus, exploiting or reversing T-cell exhaustion can be a double-edged sword in future efforts of targeting pancreatic cancer.

## Figures and Tables

**Figure 1 cancers-12-02274-f001:**
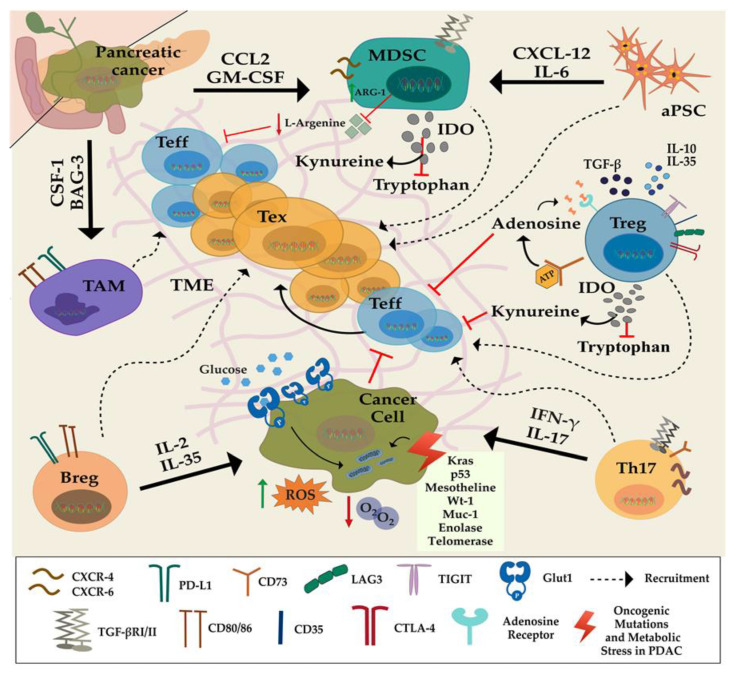
Cellular and molecular immunomodulatory factors of T-cell exhaustion in pancreatic cancer in the tumor microenvironment: myeloid-derived suppressor cells (MDSCs) and tumor-associated macrophages (TAMs) inhibit T-cell function directly and indirectly through tumor-derived proteins, such as Granulocyte-macrophage colony-stimulating factor (GM-CSF), C-C Motif Chemokine Ligand 2 (CCL2), Colony Stimulating Factor 1 (CSF1), and Bcl2-associated athanogene 3 (BAG3). Activated pancreatic stellate cells (aPSCs) recruit suppressive immune cells and impair antitumor cells in the stroma and, via secretion of interleukin 6 (IL-6) they induce immune checkpoints on T cells in a C-X-C motif chemokine 12(CXCL12)-dependent manner. They also promote the proliferation of MDSCs and IL-35 secreting Bregs. Intratumoral Tregs secrete suppressive cytokines IL-10, IL-35, tumor growth factor β (TGF-β), thereby inducing T-cell dysfunction to impair Teff cell proliferation. Tregs also elevate kynurenine concentration and reduce available tryptophan required for effector Tcell’ effector function in TME by producing indoleamine 2-3 deoxygenase (IDO). l-arginine level, which is associated with improved antitumor activity, is diminished in tumor microenvironment (TME), leading to decreased T-cell survival. Th17 cells suppress Treg function, and the role of IL-17 produced by Th17 cells is controversial. The cancer cells bearing mutations in KRAS, enolase, mesothelin in TME also contribute to T-cell dysfunction through inducing checkpoints on T cells, leading them into exhausted phenotype. Oncogene Kirsten Rat Sarcoma (KRAS) upregulates expression of GLUT-1 gene in cancer cells to increase glucose influx for glycolysis known as Warburg effect. Due to mitochondrial dysfunction, reactive oxgen species (ROS) level is increased in pancreatic cancer cells, which promotes tumor progression.

**Figure 2 cancers-12-02274-f002:**
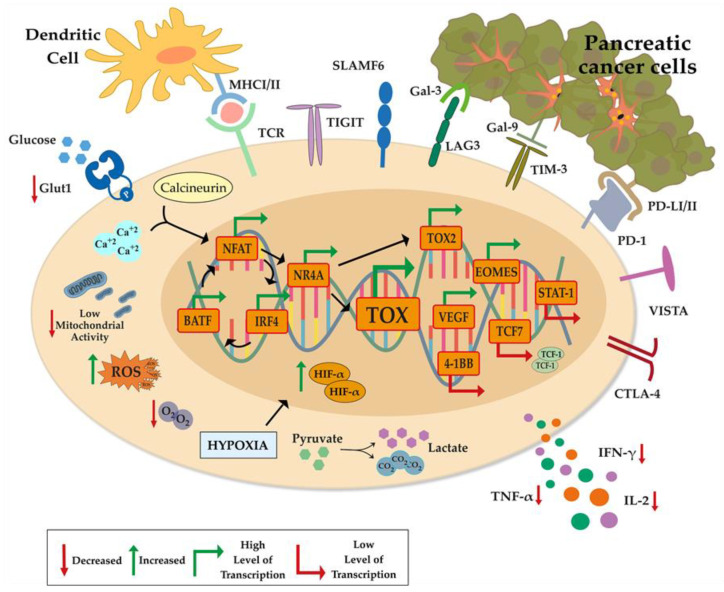
Epigenetic and transcriptional reprogramming in an exhausted T cell. Exhausted T cells have many immune checkpoints (CTLA-4, PD-1, LAG-3, TIM-3, TIGIT, SLAMF6, and VISTA) that are directly or indirectly induced by the tumor and other cells in TME. Aerobic glycolysis in the Tex cytoplasm is limited by the down-regulation of GLUT-1, which is activated by the binding of glucose. As a result of the calcineurin cycle, it causes high expression on NFAT and TCR genes. Exhausted T cells have low activity of mitochondrial function. The resulting mitochondrial activity decrease induces ROS production. As a result of hypoxia, which plays an essential role in tumor survival and Tex, it causes an increased HIFα level and increases lactate and CO2 with degraded pyruvate. A decrease in IFN-γ, TNF-α, and cytokine secretion, which play an essential role in T-cell activity, is observed when driven to exhaustion. In the exhausted T-cell nucleus; NFAT, BATF and IRF4 genes exhibit high expression levels by inducing each other. This gene feedback induces TOX genes, which play an important role in NFAT, NR4A, and T-cell exhaustion. While Vegf, Tox2, and Eomes genes, which are dependent on TOX gene expression, are more induced in their expression, Stat-1, 4-1BB, and Tcf7 genes are observed at a low level. (PD-1: Programmed cell death protein-1, CTLA-4: cytotoxic T-lymphocyte-associated protein 4, TIM-3: T cell immunoglobulin and mucin domain-containing protein 3, Lag-3: Lymphocyte-activation gene 3, SLAMF6: Self-ligand receptor of the signaling lymphocytic activation molecule, VISTA: V-domain Ig suppressor of T cell activation, GLUT-1: Glucose transporter 1, IFN-γ: Interferon gamma-γ, TNF -α: Tumor necrosis factor alpha, TOX: Thymocyte Selection Associated High Mobility Group Box).

**Table 1 cancers-12-02274-t001:** Clinical trials of potential immunotherapeutic targeting main or mediatory immunosuppressive molecules for PDAC treatment.

Immune Target Category	Immune Target	Clinical Trial Number	Medication Name	Results and Comments
Inhibitory Receptor and Ligands	*PD-1*	NCT02009449 NCT02526017 NCT02423954 NCT02451982 NCT03161379 NCT03214250 NCT03190265	Nivolumab(BMS-936558/MDX-1106/ONO-4538)	No objective response with single PD-1 blockade, partly effective with patients of MSI-high tumors (FDA approved). Combinations with other immunotherapies such as GVAX vaccine, chemotherapies or radiotherapies are still under investigation.
		NCT02648282 NCT02362048 NCT02305186 NCT02546531 NCT02432963 NCT01174121 NCT03331562	Pembrolizumab(MK-3475/SCH 900475)	
		NCT01313416 NCT01386502	Pidilizumab (CT-011)	
	*PD-L1*	NCT02669914 NCT02527434 NCT02586987 NCT02558894 NCT02583477 NCT02301130 NCT02639026	Durvalumab (MEDI4736)	
	*CTLA-4*	NCT01473940 NCT01896869 NCT03190265 NCT02527434 NCT02558894 NCT02301130 NCT02639026	Ipilimumab (BMS-734016/MDX-010) Tremelimumab (CP-675/CP-675,206)	
Effector Receptor	CD40	NCT02588443	RO7009789 (anti-CD40)	May benefit from combination with checkpoint inhibitors
	CD137	NCT02451982	Urelumab	
	CD20	NCT00001805	Rituximab	
TME Targeting Agents	IDO	NCT02077881	Indoximod	Studies are still in Phase 1 and Phase 2. No objective result used mostly with combination.
	BTK	NCT02403271	Ibrutinib	
	CCR2/CCR5	NCT03767582	CCR2/CCR5 dual antagonist (BMS-813160)	
	TGF-ß	NCT00844064	AP 12009 (trabedersen)	
Therapeutic Vaccines	GM-CSF	NCT01417000 NCT02004262 NCT00084383 NCT00305760 NCT00836407	GVAX	No objective result with single vaccine, combination with checkpoint inhibitors improves survival. Phase 1 study of ipilimumab with GVAX vaccination showed prolonged survival and improved anti-cancer T-cell response
	All immune cells	NCT01072981 NCT00569387 NCT00255827	Algenpantucel-L	
	Telomerase peptide	NCT00425360 NCT01342224	GV1001	
	MUC1	NCT00008099	MUC1antigen/SB AS-2	
	WT1	NCT03114631	MUC-1/WT-1peptide	

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
