# Peer review of "Mechanisms of T-Cell Exhaustion in Pancreatic Cancer"

_cancers, 2020, doi:10.3390/cancers12082274_

Round 1
Reviewer 1 Report
Saka et al. provide a comprehensive overview of the T cell exhaustion in PDAC. The authors focus on very interesting topics in T cell exhaustion eg. the tumor microenvironment and transcriptional and epigenetic changes. The immunotherapy section is also well structured and comprehensive. This review provides an informative overview and valuable insight into T cell exhaustion researches and their clinical implications which is quite valuable to the community. I have a few suggestions before I would happily accept this review.
For Immunosuppressive Cells, the authors mentioned Regulatory T cells (Tregs), tumor-associated macrophages (TAMs), Myeloid-derived suppressor cells (MDSCs) and regulatory B cells (Bregs). However, only Tregs is sufficiently discussed. It would be great if the authors can elaborate more on the effects of other mentioned cell types.
It seems to me that Inhibitory Receptors, CTLA-4, PD-1/PDL-1, LAG-3, Galectin Family, TIM3 and etc are the characterization of exhausted T cell rather than the tumor microenvironment factors. It would be better to discuss them in a new section.
please cite relevant papers for the following claims.
Line 64 to 70
Line 117 to 120
Line 155
Line 175-176
Line 187-188
Line 197-199
Line 208-209
Line 245-247
Line 267-268
Line 289-291
Line 301-304
Line 349-352
Line 449-451
Line 477-479
Line 481-482
Line 509-510
Line 532-535
Line 540-541
Author Response
Response to reviewer 1:
Saka et al. provide a comprehensive overview of the T cell exhaustion in PDAC. The authors focus on very interesting topics in T cell exhaustion eg. the tumor microenvironment and transcriptional and epigenetic changes. The immunotherapy section is also well structured and comprehensive. This review provides an informative overview and valuable insight into T cell exhaustion researches and their clinical implications which is quite valuable to the community. I have a few suggestions before I would happily accept this review.
1-For Immunosuppressive Cells, the authors mentioned Regulatory T cells (Tregs), tumor-associated macrophages (TAMs), Myeloid-derived suppressor cells (MDSCs) and regulatory B cells (Bregs). However, only Tregs is sufficiently discussed. It would be great if the authors can elaborate more on the effects of other mentioned cell types.
- ANSWER: TAMs and MDSCs: As the reviewer suggested, we have extended ‘immunosuppressive cells’ section and included studies on TAMs role in PDAC between Line 128-146. We have also mentioned how the location of TAMs influences T cell immunity in PDAC and its importance as well as immunosuppressive soluble molecules that inhibit effector function of T cells secreted by TAMs. Similarly, we elaborated more on the PDAC studies that discuss MDSCs infiltration and MDSC- and TAM-derived soluble molecules causing amino acid depletion in the TME of PDAC.
A paragraph with 3 references is added in to the main text about Bregs
Here are the lines newly added into the main text:
Line 105-106: IL-17 is a potent cytokine which induces the stimulation of such as IL-6, TNF, G-CSF, chemokines, and matrix metalloproteases to induce inflammation [39]
Line 126: ……Th17, suggesting that IL-6 promotes this shift in TGF-β- rich pancreatic TME.
Line 128- 146: M2 type anti-inflammatory macrophages called TAMs also play a significant role in pancreatic tumor progression and metastasis by facilitating immunosuppressive environment for antitumor T cells activity and proliferation through induction of immunosuppressive cytokines and enhancing the immunosuppressive capacity and the number of tumor stem-like cells in PDAC [46]. In general, TAMs, once activated by Th2 cytokines, use many strategies to induce immunosuppression. They secrete suppressive cytokines and factors, IL-10, IL-35, and TGF-β, which contribute to the impairment of Teff proliferation and activity[47]. Alternatively, TAMs can induce exhaustion by inducing PD-L1 expression on monocytes, which bind to PD-1 on CD8+ cells. Besides, they can also inhibit Teff activity by producing enzymes that deplete certain amino acids in the environment needed for Teff metabolism. As such, the overexpression of CD73 and CD39 ectoenzymes by TAMs generate pericellular adenosine and cause suppression of Teff via activation of the adenosine A2A receptor and eventually cause apoptosis [48] (Fig. 2). Therefore, the modulation of TAMs has been of great interest in recent years to overcome exhaustion and dysfunction of T cells and to achieve significant antitumor responses in therapies. Zhu et al, showed that the blockade of CSF/CSFR1 signaling significantly decreased the number of tumor-infiltrating TAMs and led to the reprogramming of TAMs, which produce less immunosuppressive and more antitumorigenic factors. Interestingly, CSF1/CSFR1 blockade achieved up to 85% tumor regression in a murine model when combined with PD1/CTLA4 inhibitors and gemcitabine, improved tumor regression in this murine model as well as increased effector CD8+ and CD4+ TIL infiltration and activity [49].
Line 151-153: MDSCs are immature myeloid cells, which suppress antitumoral immunity, leading to cancer progression. There are two major types: 1) the predominant one in PDAC cells: Polymorphonuclear (PMN-MDSCs) and 2) mononuclear (M-MDSCs) [52].
Line 154-156: It is shown that depletion of a single myeloid subset, the G-MDSC, can unmask an endogenous T cell response, revealing an unexpected latent immunity in GEMM of PDAC [53]
Line 164-168: Zhu et al., define two main subsets of macrophages in PDAC, 1) monocyte-derived and 2) tissue-resident TAMs. Tissue-resident TAMs not only persisted but undergo significant expansion during PDAC progression. They also showed that tissue-resident TAMs are more important for progression than monocyte-derived TAMs since having higher pro-fibrotic profile and their depletion significantly reduced tumor progression [57].
Line 173-177: A subset of B cells called Bregs are demonstrated to have immunoregulatory functions through secretion of tolerogenic cytokines such as TGF-β and IL-10 [61]. Guo et al. detected high IL-18 level in pancreatic cancer patients [62].Furthermore, IL-18 is found to be responsible for the immunosuppression and decreased Teff activity in pancreatic cancer via inducing Breg proliferation, which then leads upregulation of PD-1 receptor in B cells [63].
2-It seems to me that Inhibitory Receptors, CTLA-4, PD-1/PDL-1, LAG-3, Galectin Family, TIM3 and etc are the characterization of exhausted T cell rather than the tumor microenvironment factors. It would be better to discuss them in a new section.
- ANSWER: As it was suggested by the reviewer, we discussed “ Inhibitory Receptors” as a separate section (SECTION 3) starting from Line 228
3-please cite relevant papers for the following claims.
- ANSWER:
Line 64 to 70 became 68-74 : Ref [17-20] were added
- Feig, C., et al., The pancreas cancer microenvironment. Clin Cancer Res, 2012. 18(16): p. 4266-76.
- Demir, I.E., H. Friess, and G.O. Ceyhan, Neural plasticity in pancreatitis and pancreatic cancer. Nat Rev Gastroenterol Hepatol, 2015. 12(11): p. 649-59.
- Ozdemir, B.C., et al., Depletion of Carcinoma-Associated Fibroblasts and Fibrosis Induces Immunosuppression and Accelerates Pancreas Cancer with Reduced Survival. Cancer Cell, 2015. 28(6): p. 831-833.
- Neesse, A., et al., Stromal biology and therapy in pancreatic cancer: a changing paradigm. Gut, 2015. 64(9): p. 1476-84.
Line 117 to 120: became 128-139: Ref [46-48] were added.
- Mitchem, J.B., et al., Targeting tumor-infiltrating macrophages decreases tumor-initiating cells, relieves immunosuppression, and improves chemotherapeutic responses. Cancer Res, 2013. 73(3): p. 1128-41.
- Liu, S., et al., Molecular and clinical characterization of CD163 expression via large-scale analysis in glioma. Oncoimmunology, 2019. 8(7): p. 1601478.
- Beavis, P.A., et al., Adenosine Receptor 2A Blockade Increases the Efficacy of Anti-PD-1 through Enhanced Antitumor T-cell Responses. Cancer Immunol Res, 2015. 3(5): p. 506-17.
Line 155 became 201 : Ref [72-73] were added
- Uyttenhove, C., et al., Evidence for a tumoral immune resistance mechanism based on tryptophan degradation by indoleamine 2,3-dioxygenase. Nat Med, 2003. 9(10): p. 1269-74.
- Geiger, R., et al., L-Arginine Modulates T Cell Metabolism and Enhances Survival and Anti-tumor Activity. Cell, 2016. 167(3): p. 829-842 e13.
Line 175-176 revised and edited became 221-225 Ref [88] was added
- Zhou, L., et al., The distinct role of CD73 in the progression of pancreatic cancer. J Mol Med (Berl), 2019. 97(6): p. 803-815.Line 187-188
Line 187-188 these lines are removed from the text.
Line 197-199 became 251-254: Ref [91,98-103, 15] were added
- Ahmadzadeh, M., et al., Tumor antigen-specific CD8 T cells infiltrating the tumor express high levels of PD-1 and are functionally impaired. Blood, 2009. 114(8): p. 1537-44.
- Sfanos, K.S., et al., Human prostate-infiltrating CD8+ T lymphocytes are oligoclonal and PD-1+. Prostate, 2009. 69(15): p. 1694-703.
- Keir, M.E., et al., PD-1 and its ligands in tolerance and immunity. Annu Rev Immunol, 2008. 26: p. 677-704.
- Agata, Y., et al., Expression of the PD-1 antigen on the surface of stimulated mouse T and B lymphocytes. Int Immunol, 1996. 8(5): p. 765-72.
- Ohaegbulam, K.C., et al., Human cancer immunotherapy with antibodies to the PD-1 and PD-L1 pathway. Trends Mol Med, 2015. 21(1): p. 24-33.
- Hui, E., et al., T cell costimulatory receptor CD28 is a primary target for PD-1-mediated inhibition. Science, 2017. 355(6332): p. 1428-1433.
- Rahn, S., et al., POLE Score: a comprehensive profiling of programmed death 1 ligand 1 expression in pancreatic ductal adenocarcinoma. Oncotarget, 2019. 10(16): p. 1572-1588.
- Wherry, E.J. and M. Kurachi, Molecular and cellular insights into T cell exhaustion. Nat Rev Immunol, 2015. 15(8): p. 486-99.
Line 208-209 became 269-270: Ref [110] was added
- Andrews, L.P., et al., LAG3 (CD223) as a cancer immunotherapy target. Immunol Rev, 2017. 276(1): p. 80-96.
Line 245-247 became 305: Ref [126] was added
- Levin, S.D., et al., Vstm3 is a member of the CD28 family and an important modulator of T-cell function. Eur J Immunol, 2011. 41(4): p. 902-15.
Line 267-268 became 326-327: Ref [129-130] was added
- Cocks, B.G., et al., A novel receptor involved in T-cell activation. Nature, 1995. 376(6537): p. 260-3.
- Veillette, A. and S. Latour, The SLAM family of immune-cell receptors. Curr Opin Immunol, 2003. 15(3): p. 277-85.
Line 289-291 became 346: Ref: [136] was added
- Wolf, Y., A.C. Anderson, and V.K. Kuchroo, TIM3 comes of age as an inhibitory receptor. Nat Rev Immunol, 2020. 20(3): p. 173-185.
Line 301-304 became 365-368: Ref [13,142-145] were added
- Bengsch, B., et al., Epigenomic-Guided Mass Cytometry Profiling Reveals Disease-Specific Features of Exhausted CD8 T Cells. Immunity, 2018. 48(5): p. 1029-1045 e5.
- Scott, A.C., et al., TOX is a critical regulator of tumour-specific T cell differentiation. Nature, 2019. 571(7764): p. 270-274.
- Khan, O., et al., TOX transcriptionally and epigenetically programs CD8(+) T cell exhaustion. Nature, 2019. 571(7764): p. 211-218.
- Seo, H., et al., TOX and TOX2 transcription factors cooperate with NR4A transcription factors to impose CD8(+) T cell exhaustion. Proc Natl Acad Sci U S A, 2019. 116(25): p. 12410-12415.
- Alfei, F., et al., TOX reinforces the phenotype and longevity of exhausted T cells in chronic viral infection. Nature, 2019. 571(7764): p. 265-269.
Line 349-352 became 376-380: Ref [146] was added
- Alfei, F., et al., TOX reinforces the phenotype and longevity of exhausted T cells in chronic viral infection. Nature, 2019. 571(7764): p. 265-269.
Line 449-451 became 489-491: Ref [164] was added
- Man, K., et al., Transcription Factor IRF4 Promotes CD8(+) T Cell Exhaustion and Limits the Development of Memory-like T Cells during Chronic Infection. Immunity, 2017. 47(6): p. 1129-1141 e5.
Line 477-479 became 543-544: Ref [175] was added
- Jacobs, S.R., et al., Glucose uptake is limiting in T cell activation and requires CD28-mediated Akt-dependent and independent pathways. J Immunol, 2008. 180(7): p. 4476-86.
Line 481-482 is revised and edited became 546-548 : Ref [177] was added
- Chang, C.H., et al., Posttranscriptional control of T cell effector function by aerobic glycolysis. Cell, 2013. 153(6): p. 1239-51.
Line 509-510 became 569-571 Ref: [3,54,69,184] were added
- Binnewies, M., et al., Understanding the tumor immune microenvironment (TIME) for effective therapy. Nat Med, 2018. 24(5): p. 541-550
- Bayne, L.J., et al., Tumor-derived granulocyte-macrophage colony-stimulating factor regulates myeloid inflammation and T cell immunity in pancreatic cancer. Cancer Cell, 2012. 21(6): p. 822-35.
- Ene-Obong, A., et al., Activated pancreatic stellate cells sequester CD8+ T cells to reduce their infiltration of the juxtatumoral compartment of pancreatic ductal adenocarcinoma. Gastroenterology, 2013. 145(5): p. 1121-32.
- Peranzoni, E., et al., Macrophages impede CD8 T cells from reaching tumor cells and limit the efficacy of anti-PD-1 treat PD-1 treatment. Proc Natl Acad Sci U S A, 2018. 115(17): p. E4041-E4050.
Line 532-535 became 599-601:Ref [193] was added
- Le, D.T., et al., Evaluation of ipilimumab in combination with allogeneic pancreatic tumor cells transfected with a GM-CSF gene in previously treated pancreatic cancer. J Immunother, 2013. 36(7): p. 382-9.
Line 540-541 became 609-612: Ref [195] was added
- Paley, M.A., et al., Progenitor and terminal subsets of CD8+ T cells cooperate to contain chronic viral infection. Science, 2012. 338(6111): p. 1220-5.
Reviewer 2 Report
In this review, Saka & Gokalp et al. describe emerging mechanisms of T-cell exhaustion in pancreatic cancer with a particular emphasis on the role of these events in immune checkpoint inhibition. Overall, this review is systematic, comprehensive, and well thought out. I have only minor suggestions on how to improve it.
The Vonderheide group has performed extensive preclinical and clinical work pertaining to CD40 agonists in PDAC. Though these are included in Table form, this review would benefit from a deeper discussion pertaining to the mechanisms though which CD40 agonists may overcome T-cell exhaustion in PDAC, though the toxicity of these approaches warrants mention (Grewal et al., Nature 1995; Vonderheide et al., JCO 2007; Beatty et al., Science 2011; Beatty et al., Clin Cancer Res 2013; O'Hara et al., Cancer Res 2019;79(13 Supplement):CT004-CT04).
Additionally, the authors should include mention of newly emerging basic science and clinical studies that are exploring selected mechanisms described in this work including the addition of select inhibitors of many targets mentioned in this study. These include but are not limited to: PEGPH20 (Chiorean et al., JCO 2020), CXCR4 (Hidalgo et al.,Annals of Oncology 2019;Volume 30(Supplement 11), and TGF-beta (Strauss et al., Clin Can Res 2018; Principe et al., Cancer Res 2020; Principe et al., Mol Cancer Therapeutics 2019; Melisi et al., JCO 2019;37(15_suppl):4124-24).
Author Response
Response to Reviewer 2
In this review, Saka & Gokalp et al. describe emerging mechanisms of T-cell exhaustion in pancreatic cancer with a particular emphasis on the role of these events in immune checkpoint inhibition. Overall, this review is systematic, comprehensive, and well thought out. I have only minor suggestions on how to improve it.
-The Vonderheide group has performed extensive preclinical and clinical work pertaining to CD40 agonists in PDAC. Though these are included in Table form, this review would benefit from a deeper discussion pertaining to the mechanisms though which CD40 agonists may overcome T-cell exhaustion in PDAC, though the toxicity of these approaches warrants mention (Grewal et al., Nature 1995; Vonderheide et al., JCO 2007; Beatty et al., Science 2011; Beatty et al., Clin Cancer Res 2013; O'Hara et al., Cancer Res 2019;79(13 Supplement):CT004-CT04).
- ANSWER: We discuss the papers that the reviewer suggested, in the main text and included the paragraph below into main text.
Line: 618-626: Activation of the CD40 receptors by tumor cells is an important step for T cell immunity. Thus, it is worthy of mentioning that stimulating antigen-presenting cells is also a way to boost the immune responses in PDAC. CD40 agonist mAb is one of the possible targets, which has been shown to have antitumor activity in solid malignancies. [198, 199]. In the case of PDAC, the clinical trial of CD40 agonist mAb combined with gemcitabine gave hopeful results [200]. Also, the phase 1 study CD40 agonistic mAb plus gemcitabine and nab-paclitaxel with or without nivolumab showed significant antitumor activity in PDAC patients [201]. Although these studies gave promising results, their toxicity such as cytokine release syndrome, vascular and hematologic complications, and liver toxicity should be kept in mind.
-Additionally, the authors should include mention of newly emerging basic science and clinical studies that are exploring selected mechanisms described in this work including the addition of select inhibitors of many targets mentioned in this study. These include but are not limited to: PEGPH20 (Chiorean et al., JCO 2020), CXCR4 (Hidalgo et al.,Annals of Oncology 2019;Volume 30(Supplement 11), and TGF-beta (Strauss et al., Clin Can Res 2018; Principe et al., Cancer Res 2020; Principe et al., Mol Cancer Therapeutics 2019; Melisi et al., JCO 2019;37(15_suppl):4124-24).
- ANSWER: To the immunotherapy of PDAC (section 6), we included the suggested studies and explained the clinical outcomes briefly.
Line 597-601: Due to limited effect of PD-1 inhibitors in pancreatic cancer, it is shown that CXC chemokine receptor 4 (CXCR4) blockade promotes T cell infiltration with its synergistic effect with PD-1 inhibitors in mouse models[67, 192]. Bockorny et al. also showed that combined CXCR4 and PD-1 inhibitors promoted an increase in T cell infiltration and decrease in MDSCs in pancreatic cancer patients [193].
Line 627: Inhibitors of TGF-β were shown effective in preclinical models [206] and combination of TGF-β inhibitors with gemcitabine improved overall survival compared to gemcitabine alone in patients with unresectable pancreatic cancer [207].
Note: We decided not to include and discuss clinical trials on PEGHP20 in our review due to very recent outcomes of PEGPH20 clinical trials that either PEGHP20 and pegylated interleukin-10 (pegilodecakin) failed to improve overall survival of pancreatic cancer patients.
Reviewer 3 Report
This manuscript provides a review of known and potential immunosuppressive mechanisms in pancreatic cancer. It is overall reasonably well written, but there are a few typos and incomplete sentences.
Much of the manuscript could be cut down in length by removing unsupported statements and sections that are only loosely connected to pancreatic cancer.
Some examples of areas of concern:
The introduction has incorrect numbers of patients diagnosed with and dying of pancreatic cancer in the US (line 30; off by an order of magnitude).
There is repeated reference to pancreatic cancer having low immunogenicity and poor neoantigen presentation (line 34), without strong evidence. Often, references for specific statements are to review papers, rather than primary literature.
Expression of CTLA-4, PD-1, and other checkpoint receptors do not necessarily prevent T cell activation, in the absence of cognate ligand binding (lines 182-185).
Describing PDAC as immune quiescent is not completely accurate (line 202 and elsewhere). Review of broad literature on human disease will show that there is a large immune infiltrate including effector T cells.
Author Response
Response to Reviewer 3
This manuscript provides a review of known and potential immunosuppressive mechanisms in pancreatic cancer. It is overall reasonably well written, but there are a few typos and incomplete sentences.
- ANSWER: Grammatical errors and typos are corrected.
Much of the manuscript could be cut down in length by removing unsupported statements and sections that are only loosely connected to pancreatic cancer.
- ANSWER: We thank the Reviewer for this comment. In the revised version, we integrated additional references and statements (see also above) into the text to avoid unsupported statements.
Some examples of areas of concern:
-The introduction has in numbers of patients diagnosed with and dying of pancreatic cancer in the US (lincorrect e 30; off by an order of magnitude).
- ANSWER:
Line 30: the sentence was revised to “Globally, the mortality numbers are very close to incidence numbers projecting pancreatic cancer as the 7th leading cause of cancer -related deaths[1]. Globocan statistics predict the incidence number to be almost doubled by 2040 (http://globocan. iarc.fr/) [1].
-There is repeated reference to pancreatic cancer having low immunogenicity and poor neoantigen presentation (line 34), without strong evidence. Often, references for specific statements are to review papers, rather than primary literature .
- ANSWER: Binneweis et al was cited for poor immunogenic characteristics of tumor environment, which exists in PDAC (defined as ‘broadly populated with immune cells but are relatively void of CTLs in the tumor core which is called infiltrated-excluded tumor immune microenvironement (I-E) type’). Also, original research papers have been provided referring to PDAC’s poor neoantigen presentation and low immunogenicity as shown in the paragraphs below and in the main text. On the contrary, we agree that our statement related to the unresponsiveness of PDAC to immunotherapy due to low tumor antigenicity and due the upregulation of IRs is quite too strong to claim. Therefore, we have removed this sentence “Due to low tumor antigenicity and upregulation of IRs stimulated by the immunosuppressive TME, PDAC has not been proven to be adequately responsive to immunotherapies, unlike other cancers.” has been removed from the main text. The corresponding citations to the statements in the following paragraphs have been included in the main text and some paragraphs are revised and paraphrased accordingly.
In the following paragraphs, both original and review papers were cited.
Line 34-39: PDAC tumors are unresponsive and or mildly responsive o chemotherapy, radiotherapy, and immunotherapy. The properties of desmoplastic dense stroma impairing drug delivery [3], relatively low mutational loads and the low number of tumor neoantigens[4] [5], the poor tumor immunogenicity [6, 7], acquired tumor intrinsic therapy resistance, genetic and epigenetic instability and unique immunosuppressive tumor microenvironment (TME) are the proposed characteristics for the low therapy response.
Line 257-267: PDAC is described as 'immunologically cold’ ' compared to highly immunogenic melanoma because of very low surface presentation of neoantigens, and the insufficient Tc infiltration into the tumor core because of fibrotic trap and TAMs localized in the surrounding of tumor [3, 55, 103, 105] which result in poor clinical outcomes from immune-checkpoint inhibitors targeting PD-1/PD-L1 and CTLA-4[70, 106, 107]. A comprehensive retrospective study on resected PDAC tumors reported four major subclasses of tumors based on genomic, transcriptomic and, clinicopathological data. High levels of tumor neoantigens exist in the subtypes with impaired double-strand break and mismatched repair mechanisms, implicating that immunotherapy can be successful if applied to the right patient [108]. Fortunately, recent advances in genomics and transcriptomics have been discovering new target proteins that can improve tumor regression when combined with existing therapies for PDAC [109].
-Expression of CTLA-4, PD-1, and other checkpoint receptors do not necessarily prevent T cell activation, in the absence of cognate ligand binding (lines 182-185).
- ANSWER: The statement has been corrected and revised to ;
Line 231-232 Upon binding to their cognate ligands on cancer cells, T cells’ effector function and proliferation are gradually reduced.
Review of broad literature on human disease will show that there is a large immune infiltrate including effector T cells.
- ANSWER: The reviewer might be right at one point that there are studies showing immune infiltrates to tumor. However, PDAC is called “ a cold tumor” due to many reasons. First of all, most of the infiltrates comprise of different types of immunosuppressive cells which we discussed in the review. Secondly, highly immunosuppressive TME of PDAC suppresses CD8+ and CD4+ TILs anti-tumor function, leading exhaustion, anergy, and senesence. On top of everything, there are many studies in the literature which show that the effector T cells do not exist in the core of the primary PDAC tumor like in melanoma. Instead, they locate in the periphery because they are trapped by fibrotic environment and TAMs, MDSCs in the surrounding of tumor which we mention in line 257-260. Moreover, effector T cells in the mentioned studies have in many instances not been tested for their activation status, e.g. by checking the levels of Granzyme B expression in these cells.
Round 2
Reviewer 3 Report
Although the authors continue to argue that PDAC is "a cold tumor", the majority of my concerns have been addressed.
This manuscript is a resubmission of an earlier submission. The following is a list of the peer review reports and author responses from that submission.